# In situ edge engineering in two-dimensional transition metal dichalcogenides

Xiahan Sang [1], Xufan Li[1,4], Wen Zhao [2], Jichen Dong[2], Christopher M. Rouleau[1], David B. Geohegan[1], Feng Ding[2,3], Kai Xiao[1] & Raymond R. Unocic [1]

Exerting synthetic control over the edge structure and chemistry of two-dimensional (2D) materials is of critical importance to direct the magnetic, optical, electrical, and catalytic properties for specific applications. Here, we directly image the edge evolution of pores in $Mo_{1−x}W_xSe_2$ monolayers via atomic-resolution in situ scanning transmission electron microscopy (STEM) and demonstrate that these edges can be structurally transformed to theoretically predicted metastable atomic configurations by thermal and chemical driving forces. Density functional theory calculations and ab initio molecular dynamics simulations explain the observed thermally induced structural evolution and exceptional stability of the four most commonly observed edges based on changing chemical potential during thermal annealing. The coupling of modeling and in situ STEM imaging in changing chemical environments demonstrated here provides a pathway for the predictive and controlled atomic scale manipulation of matter for the directed synthesis of edge configurations in $Mo_{1−x}W_xSe_2$ to achieve desired functionality.

[1] Center for Nanophase Materials Sciences, Oak Ridge National Laboratory, Oak Ridge, TN 37831, USA. [2] Center for Multidimensional Carbon Materials (CMCM), Institute for Basic Science (IBS), Ulsan 689-798, Republic of Korea. [3] School of Materials Science and Engineering, Ulsan National Institute of Science and Technology (UNIST), Ulsan 689-798, Republic of Korea. [4] Present address: Honda Research Institute USA, Inc, Columbus, OH 43212, USA. These authors contributed equally: Xiahan Sang, Xufan Li, Wen Zhao. Correspondence and requests for materials should be addressed to F.D. (email: f.ding@unist.ac.kr) or to K.X. (email: xiaok@ornl.gov) or to R.R.U. (email: unocicrr@ornl.gov)

Unique material properties emerge as the size of the material along one dimension is decreased to the atomic level, and this concept is the cornerstone for the recent proliferation of research on two-dimensional (2D) materials[1–4]. The discontinuity represented by the one-dimensional (1D) edges of 2D materials, analogous to surfaces of three-dimensional materials, can significantly alter the local functionality. Understanding the critical role of edge structures and chemistry in 2D materials, and controlling their formation, has been used to enhance the electronic[5–7], magnetic[8, 9], nonlinear optical[10], photoluminescence[11], and catalytic[12, 13] properties. In addition, edge structures play a critical role in controlling the growth kinetics and morphological evolution in 2D materials[14–18]. The lower symmetry of binary and ternary 2D materials, such as in transition metal dichalcogenides (TMDs), present a diversity of edge structures and compositions compared to graphene with its zigzag (ZZ) and armchair (AC) edges, and exhibit different electronic structures and functionalities. Therefore, understanding edge evolution in TMDs is critical to the design of new, functional 2D TMD materials. Experimental techniques such as scanning tunneling microscopy and computational methods have been used to study edge properties in 2D materials;[9, 19, 20] however, real-time observation of the edge evolution and transformation in different chemical environments and at the atomic scale remains a challenge.

In this study, we performed in situ heating experiments using aberration-corrected scanning transmission electron microscopy (STEM) to track the edge evolution and transformation in a $Mo_{1-x}W_xSe_2$ ($x = 0.05$) monolayer, and demonstrate that by varying the local chemical environment, we can trigger formation of nano-pores terminated by different edge reconstructions during in situ heating and electron beam irradiation and form edge structures with metallic and/or magnetic properties.

## Results

### Formation of nanopores terminated by different edge structures during in situ heating.
Chemical vapor deposition (CVD)[21, 22] was used to synthesize $Mo_{1-x}W_xSe_2$ monolayers that were transferred onto a commercial, microelectromechanical system (MEMS)-fabricated in situ heating microchip platform (Supplementary Fig. 1). Monolayer flakes were characterized using a Nion UltraSTEM100 aberration-corrected STEM. The varying chemical environments on the monolayers stemmed from different degrees of carbon residue on the surface from the transfer process, ranging from areas with an apparent network of C residue (high C areas) to clean areas without noticeable C residue (Supplementary Fig. 2). Atomic-resolution high-angle annular dark field (HAADF) STEM images acquired from a clean area of the $Mo_{0.95}W_{0.05}Se_2$ monolayer showed a honeycomb crystal structure with alternating Mo/W and Se sites (Fig. 1a). Since the HAADF-STEM image intensity is proportional to the atom number[23], the brightest atoms are W confined within the material (white dashed circles). Slightly less intense are sites with two Se atoms, with still dimmer sites that are Mo atoms. Note that Se vacancies, which are typically observed in CVD-grown 2D TMD layers[22], are indicated by blue dashed circles.

The $Mo_{0.95}W_{0.05}Se_2$ monolayer flake on the microchip was heated rapidly (10 °C/s) to 500 °C and held at this temperature to allow for sufficient structural evolution and reconstruction of the edges[24–27]. In situ heating combined with electron beam irradiation is effective for creating fresh edges by etching pores in 2D crystals in vacuum[24–26]. Compared to as-synthesized edges that can be damaged, contaminated, or bent during sample preparation, freshly-created edges at pores in vacuum better

represent intrinsic edge structure at specific local chemical environment and can be advantageous for studying the thermodynamics and kinetics of chemistry-dependent edge reconstruction. After heating at 500 °C for 20 min, large faceted quasi-hexagonal or triangular shaped pores are terminated with various edge configurations developed in the $Mo_{0.95}W_{0.05}Se_2$ flake (Fig. 1b). The pores formed initially in areas with a high Se vacancy concentration (Supplementary Fig. 3). Since the energy required to move a Se atom to vacuum (5.3 eV) is much lower than that for a Mo atom (12.3 eV), as determined from density functional theory (DFT) simulations, Se vacancies were generated continuously during thermal exposure and electron beam irradiation, which resulted in continuous pore initiation and expansion. These results are consistent with the depletion of chalcogen atoms as previously reported during in situ heating of 2D TMDs[28]. Following Se loss, excess Mo atoms can react with carbon residue on the surface to form MoC clusters or nanoparticles with a NaCl-type crystal structure (Supplementary Fig. 4). The chemical environment in our experiment was the Mo chemical potential, defined as $\mu_{Mo} = \mu_{MoSe_2} - 2\mu_{Se}$, which was determined by two competing factors: heating and electron beam irradiation that continuously produced Se vacancies and increased $\mu_{Mo}$ over time[29], and carbon residue that reacted with excess Mo atoms and decreased $\mu_{Mo}$. Hence, the initial carbon residue concentration and the heating time were the two key parameters that controlled $\mu_{Mo}$ and dictated the formation of $\mu_{Mo}$-dependent edge structures.

Figure 1d–f show a series of HAADF-STEM images (extracted from movies) acquired at different heating times that reveal the edge evolution and reconstruction processes for areas with heavy, medium, and low initial carbon residue (Supplementary Movies 1-3, respectively, and Supplementary Fig. 5). The reconstructed edges are labeled based on the orientation of the edge with respect to the hexagonal lattice of the $Mo_{0.95}W_{0.05}Se_2$ (e.g., Se-oriented ZZSe and Mo-oriented ZZMo in Fig. 1c) and the outermost termination group: Se-terminated (-Se) or nanowire-terminated (-NW, Fig. 1c). For areas with heavy carbon residue (Fig. 1d), randomly aligned edges without well-defined structures formed throughout the heating process. During heating between 359 s (top, Fig. 1d) to 1213 s (bottom, Fig. 1d), through formation of large MoC particles at edges, the amount of carbon residue (framed by white dashed lines) decreased but remained after heating, thereby maintaining a low $\mu_{Mo}$ during the entire heating process. In regions with moderate carbon residue (Fig. 1e), $\mu_{Mo}$ remained low until the carbon residue was completely depleted, at which point $\mu_{Mo}$ increased with continued heating. Concomitantly, the edge structure evolved from exhibiting predominantly Se-terminated ZZMo edges (Fig. 1e top, 0 s, ZZMo-Se) to NW-configured edges (Fig. 1e bottom, 319 s, ZZSe-Mo-NW30 and ZZMo-NW). Structural analysis indicates that the terminating NWs on the edges are $Mo_6Se_6$ (Fig. 1c), with the same crystal structure as those fabricated using electron-beam sculpting[29]. The MoSe NWs were generally rotated by 0° (NW0) or 30° (NW30) along their longitudinal axis, as defined in Fig. 1c. In areas with very low carbon contamination (Fig. 1f), $\mu_{Mo}$ rose rapidly to a relatively high level during heating, as there is not sufficient carbon to react with excess Mo atoms. The NW-terminated edges almost fully encompassed the sides of the pore from the very early stage (Fig. 1f, 0 s) to the end (Fig. 1f, 225 s). Therefore, the particular edge structures that formed depended nearly exclusively on the Mo chemical potential (the amount of C contamination present on the surface), e.g., as $\mu_{Mo}$ increased, the edge morphology assumed three structural and chemical forms; random edges, Se-terminated edges, and NW-terminated edges. Moreover, and to the best of our knowledge, this is the first time

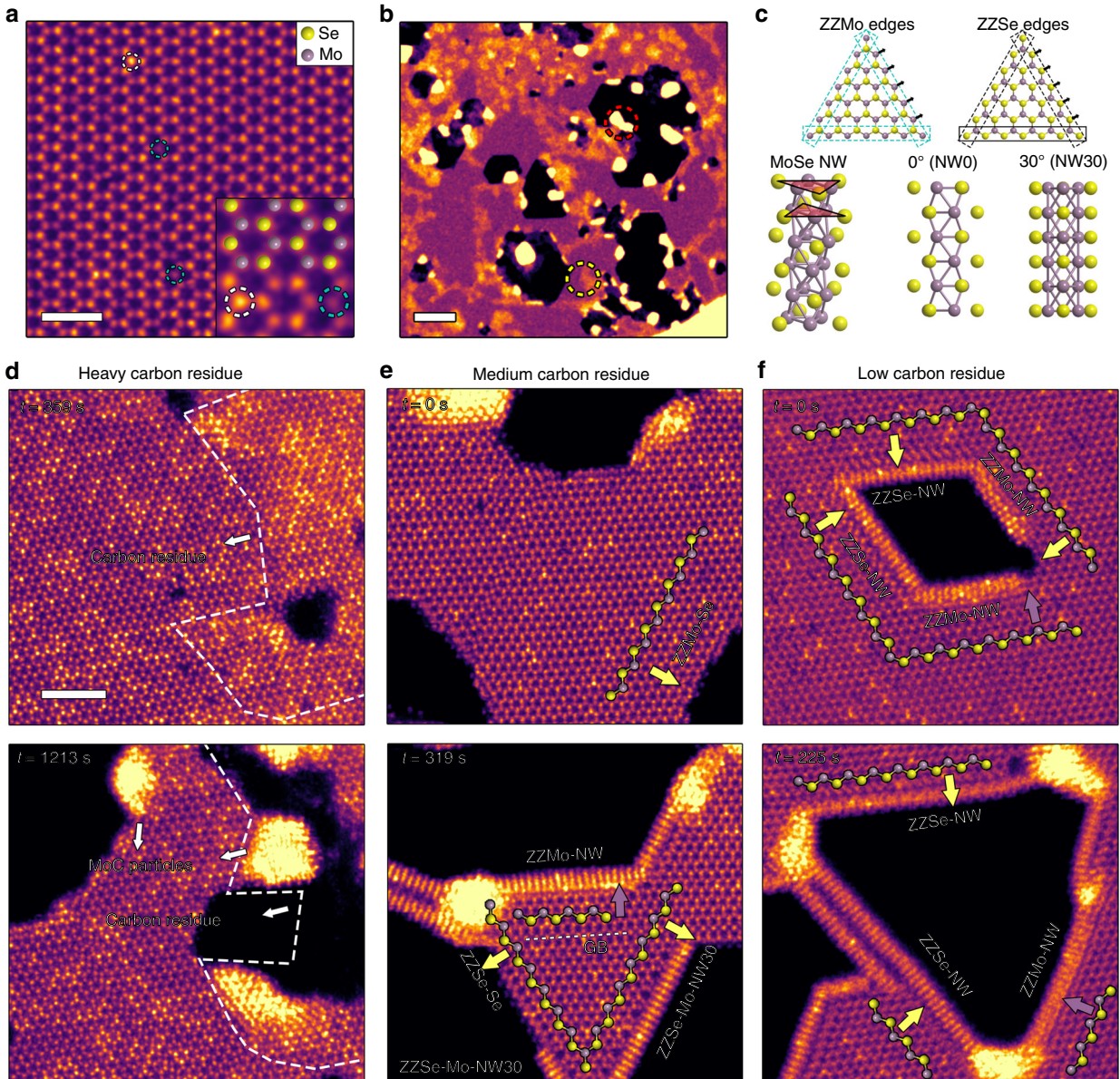

**Fig. 1** In situ heating and etching of monolayer $Mo_{0.95}W_{0.05}Se_2$ with edge reconstructions. **a** Atomic resolution HAADF-STEM image showing the crystal structure of monolayer $Mo_{0.95}W_{0.05}Se_2$. The white and cyan dashed circles indicate W dopants and Se vacancies, respectively. Inset: a structural model is overlaid on the HAADF-STEM image. Yellow, Se atoms; purple, Mo atoms. **b** Low magnification HAADF-STEM image of monolayer $Mo_{0.95}W_{0.05}Se_2$ after in situ heating at 500 °C. The red dashed circle marks a MoC particle. The yellow dashed circle indicates carbon residue area with bright contrast. **c** (Top) atomic structure models of Mo-oriented ZZMo edges with Mo atoms pointing out normal to the edge (cyan dashed boxes) and Se-oriented ZZSe edges with Se atoms pointing out of the edge (black dashed boxes). (Bottom) atomic structure models of the MoSe NW formed on the edges and the two commonly-observed projections NW0 and NW30. **d**–**f** Atomic resolution HAADF-STEM images at initial and later stages of etching from areas with heavy (**d**), medium (**e**), low (**f**) carbon residue. Carbon residue and MoC nanoparticles are indicated by white arrows. ZZSe and ZZMo edges are indicated by white and purple arrows, respectively. Scale bars are 1 nm, 10 nm, and 2 nm for (**a**), (**b**), and (**d**), respectively

that NW-terminated edges, as opposed to free-standing NWs, have been reported in 2D materials[29, 30].

**Influence of chemical potential on stable edge structures.** In contrast to the distinctive, single MoSe NW structures fabricated in ref. [29], the NWs referred to in the present work formed the edges of pores and were terminations within the monolayer, which is associated with a rearrangement of atoms at the $Mo_{0.95}W_{0.05}Se_2$—NW interface (Fig. 2a). Heating played a more important role in the formation of NW-terminated edges than electron beam irradiation since NW-terminated edges emerged in

areas far from where the electron beam was rastered during heating (Supplementary Fig. 6), and since electron beam irradiation at room temperature did not trigger their evolution[29]. During in situ heating of $Mo_{0.95}W_{0.05}Se_2$ monolayers, four stable edge structures were observed: ZZSe edges terminated with -NW (Fig. 2a) and -Se (Fig. 2b), ZZMo edges terminated with -NW (Fig. 2c), and -Se (Fig. 2d). For each edge configuration, the experimental HAADF-STEM image, an atomic structure models optimized using DFT, and the corresponding simulated HAADF-STEM image, are shown together to demonstrate the unambiguous structure correlation. The experimental and simulated HAADF-STEM images agreed well for all four edge

configurations (Fig. 2a–d, Supplementary Fig. 7). For example, the ZZSe-Mo-NW30 edge (Fig. 2a) is defined as a ZZSe edge connected via Mo(W) atoms to a NW rotated by 30°. A typical pentagon-heptagon (5|7) ring defect is reconstructed at the interface to accommodate lattice mismatch strain between the MoSe$_2$ hexagonal lattice and the edge. Similar 5|7 ring defects have been previously reported at grain boundaries in graphene and TMDs[31–33]. A twin grain boundary reconstruction is present in ZZSe-GB4-Se (Fig. 2b), where a twin grain boundary (GB4) turns the ZZSe edge into a ZZMo edge passivated by Se atoms. Such a twin grain boundary has been demonstrated to exhibit charge density wave behavior as quasi-1D metallic channels embedded in a semiconducting matrix[34, 35]. Both 5|7 ring and grain boundary reconstructions are predicted correctly by DFT. The two ZZMo edges (Fig. 2c, d) show negligible reconstruction during heating, likely due to the strong bonding between Mo atoms and -Se or -NW terminations.

To understand the edge reconstruction observed during in situ heating at 500 °C as a function of the local chemical environment, 44 ZZSe and 24 ZZMo edge configurations were constructed and their $\mu_{Mo}$-dependent formation energies were calculated using DFT ('Methods', Supplementary Note 1, Supplementary Fig. 8-15, Fig. 2e, f). DFT calculations confirmed that the four most frequently observed edges (Fig. 2a-d) along with ZZSe, ZZSe-Se, and ZZMo-Se2, as labeled in Fig. 2e, f, had the lowest formation energies within their respective $\mu_{Mo}$ window, i.e., the most stable edge structure for a specific chemical environment[36–38]. As $\mu_{Mo}$

increased, the termination group of stable ZZSe and ZZMo edges changed from -Se to -NW. Concomitantly, the concentration of Mo at the edge increased from Se-rich Se-terminated edges, to GB4-reconstructed edges, to Mo-rich NW-terminated edges. Therefore, the stability of Mo-rich edges was enhanced by increasing $\mu_{Mo}$, e.g., shifting the environment to be Mo-rich (Fig. 2e, f). A correlation between the formation of a Mo-rich twin boundary and Mo-rich environment was reported previously[34, 35]. In agreement with the theoretical predictions, NW-terminated edges were suppressed in the carbon residue-rich areas due to a low $\mu_{Mo}$ (Fig. 1d), but formed readily in carbon-free areas with high $\mu_{Mo}$ (Fig. 1f), and were delayed in areas with medium carbon residue until $\mu_{Mo}$ started to increase once the carbon atoms were depleted (Fig. 1e).

**Edge structure evolution and electronic properties.** Figures 3a–d capture the dynamic edge evolution, mediated by $\mu_{Mo}$ from a medium carbon residue area (Supplementary Movie 4). As $\mu_{Mo}$ increased with heating time, the initial ZZSe-Se edge (Fig. 3a, 0 s) transformed to a Mo-rich ZZSe-GB4-Se (Fig. 3b, 8 s), and eventually to a ZZSe-Mo-NW30 edge (Fig. 3c, d). We performed DFT and ab initio molecular dynamics (AIMD) simulations for insight on edge evolution pathways. Figure 3e shows the proposed evolution pathway from ZZSe-Se edge to ZZSe-GB4-Se edge through the addition of Mo atoms and shearing of atom rows, and the energy barriers for each step. AIMD simulations were employed

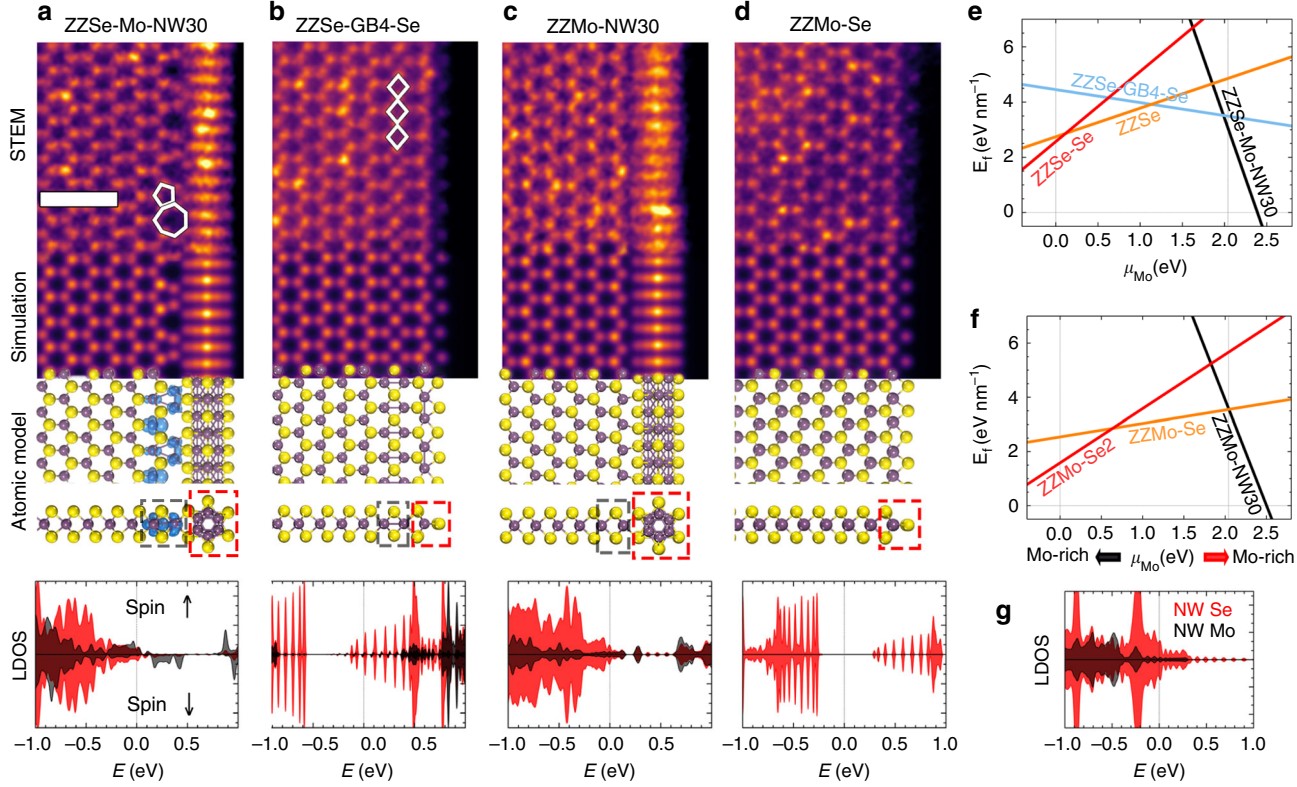

**Fig. 2** Analyses of stable edge structures formed during in situ heating. **a–d** Experimental (the 1st row) and simulated (the 2nd row) HAADF-STEM images, as well as corresponding atomic models (the 3rd row) and local density of states (LDOS) with arbitrary unit (the 4th row) of ZZSe-Mo-NW30 (**a**), ZZSe-GB4-Se (**b**), ZZMo-NW30 (**c**), and ZZMo-Se (**d**) edges, respectively. Fermi energy in LDOS is set to be 0. Red and black LDOS plots are calculated from atoms framed by red and black dashed boxes, respectively. The isosurfaces (highlighted by blue surfaces) of 0.01 e/Å$^3$ spin-magnetization density ($\Delta\rho = \rho_\uparrow - \rho_\downarrow = 0.01$ e/Å$^3$) are overlaid on the atomic structure of ZZSe-Mo-NW30 in **a**. The other three edges have negligible spin-magnetization. Scale bar corresponds to 1 nm. **e, f** DFT-calculated formation energy per unit length ($E_f$) as a function of the Mo chemical potential, $\mu_{Mo}$, for four Se-oriented (ZZSe) edges (**e**) and three Mo-oriented (ZZMo) edges (**f**). The two vertical dashed lines denote the range 0 eV < $\mu_{Mo}$ < 2.04 eV, where MoSe$_2$ can remain stable with respect to the formation of bulk Mo ($\mu_{Mo} = 2.04$ eV) or bulk α-Se ($\mu_{Mo} = 0$ eV). **g** LDOS of freestanding MoSe NW contributed by Mo (black) and Se (red) atoms

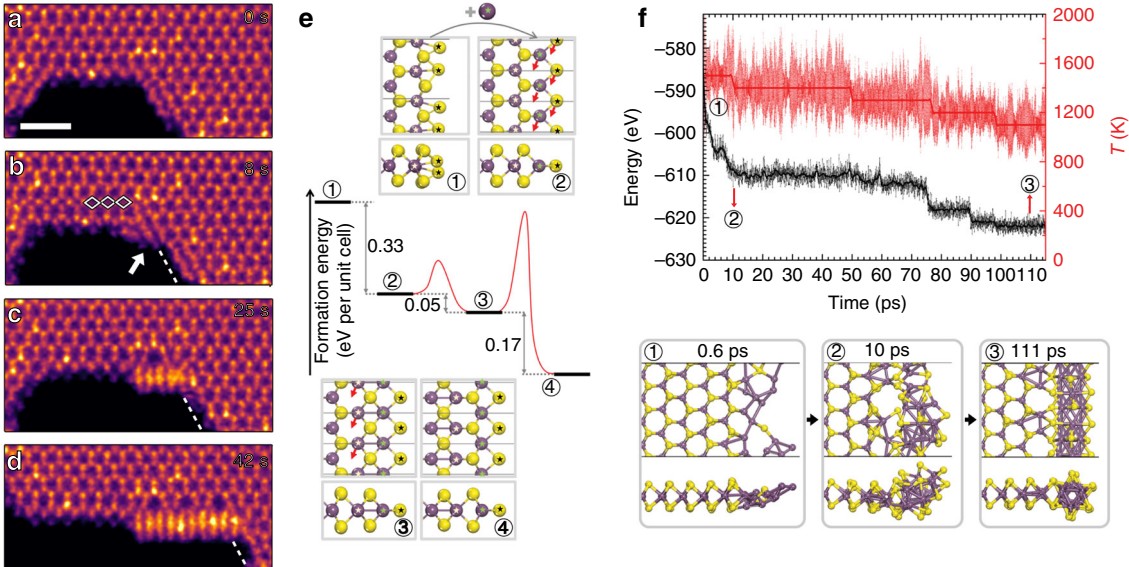

**Fig. 3** Edge reconstruction dynamics. **a–d** Atomic resolution HAADF-STEM image frames showing typical states in the process of edge reconstruction from a ZZSe-Se to ZZSe-Mo-NW30. Scale bar is 1 nm. **e** Total energy (black) and energy barrier (red) associated with the four critical steps for ZZSe-Se (**a**) to ZZSe-GB4-Se (**b**) transformation, calculated at $\mu_{Mo} = 1.2$ eV. Top and front views of the relaxed structures of the four critical steps are displayed. **f** (Top) AIMD simulation of the edge energy (black) and system temperature (red) plotted over time with a time interval of 1 fs in the process of NW evolution. The black and red solid bold lines, representing energy and temperature respectively, are smoothed data to guide the eye. The energy of the system keeps decreasing, indicating increased stability. (Bottom) top and front view snapshots of edge structure at noted times

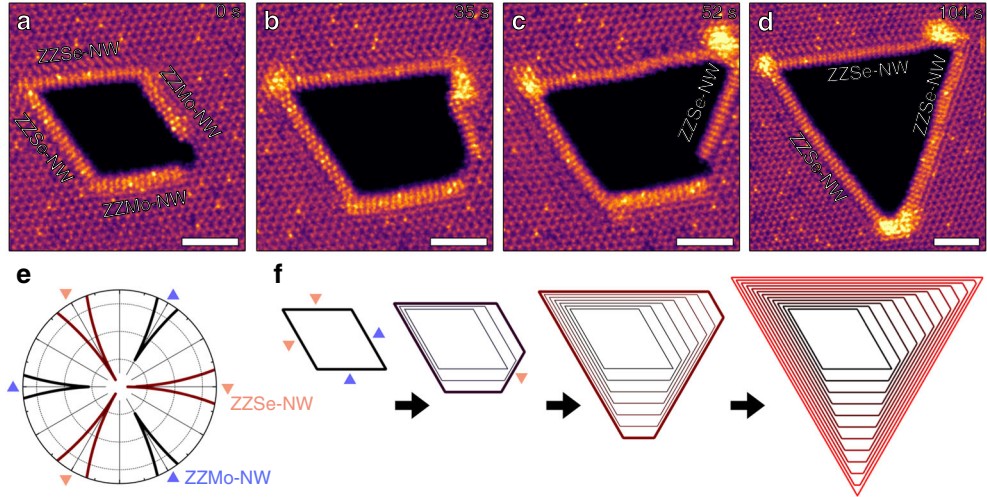

**Fig. 4** Shape evolution of an etched pore with all edges terminated by NWs. **a–d** Atomic resolution HAADF-STEM images showing the shape evolution of an initial NW-terminated parallelogram pore. **e** Polar plot of etching rates of NW-terminated ZZSe and ZZMo edge. ZZSe-NW (brown) etches more slowly than ZZMo-NW (black). **f** Simulated etching progress of the NW-terminated parallelogram pore shown in **a**. Scale bars are 2 nm

to explore the transformation from a normal edge to a NW-terminated edge, which entailed a more complicated and radical structural rearrangement (Fig. 3f). Initially, the Mo-rich environment was created by adding Mo atoms to a ZZSe edge of a MoSe₂ ribbon and the system was annealed from 1500 to 1000 K (see 'Method') over ~115 ps. The energy and temperature evolution of the system, and the typical metastable structures during the simulation, are shown in Fig. 3f (Supplementary Movie 5, Supplementary Fig. 16). AIMD simulations revealed that the NW-terminated edge with a 5|7 interface formed by the reconstruction of the Mo atoms (~10 ps) and the fast diffusion of Se atoms. Compared with the disordered NW structure, the

crystallized edge has exceptional high stability and is therefore the global minimum of the edge structure in the high-Mo-rich environment.

The stable edges at different $\mu_{Mo}$ exhibit distinctly different electronic properties and magnetic orderings. The calculated local density of states (LDOS) for the four common edges (4th row in Fig. 2a-d) indicate that the two NW-terminated edges are conductive (non-zero LDOS at the Fermi energy, $E = 0$), inherited from the metallic nature of freestanding NWs (Fig. 2g)[29]. Although both Se-terminated edges share the same atomic structure on the edge and similar LDOS contour (red dashed boxes and red LDOS in Fig. 2b, d), the ZZMo-Se edge is

semiconducting with a band gap of 0.48 eV, while the Mo-rich ZZSe-GB4-Se edge is metallic, stemming from the metallic GB4 mirror twin boundary in TMDC materials[34, 35]. Out of the four edges, the ZZSe-Mo-NW30 is the only ferromagnetic edge (Fig. 2a) as each 5|7 pair at the interface has a local magnetic moment of 0.88 $\mu_B$ due to the splitting of Mo $d_{z^2}$ and $d_{x^2-y^2}$ orbitals (Supplementary Fig. 17). Therefore, through tuning of the stable edge structures, the chemical environment can further influence edge electronic and magnetic properties, affecting the optical and transport properties of the material, which are significant for spintronic or catalytic applications.

**Pore expansion mediated by NW-terminated edges**. The edge structure directly impacts the growth and/or etching kinetics of 2D materials. While Se-terminated edges tend to be etched along the lateral direction at a fast rate, NW-terminated edges prefer to move/grow along the longitudinal direction (Supplementary Fig. 18) because lateral movement requires overcoming a high formation energy (~6 eV) to form kinks on the NW (Supplementary Fig. 19). An interesting case arises when all the edges are terminated with NWs at high $\mu_{Mo}$ (Fig. 4a–d). Over time, the initial parallelogram-shaped pore terminated with two ZZSe-NW edges and two ZZMo-NW edges (Fig. 4a) evolved into a triangular pore terminated with ZZSe-NW edges on all three sides (Fig. 4d). The two original ZZSe-NW edges grew strictly along the longitudinal direction, while the two ZZMo-NW edges consecutively disappear after 52 s (Fig. 4c) and 104 s (Fig. 4d), respectively. Using kinetic Wulff construction (KWC) theory[18], the observed ZZSe-NW dominated edge evolution can be interpreted from the different estimated etching rates of ZZMo-NW and ZZSe-NW, resulting from their different kink formation energies (5.78 eV for ZZMo-NW and 6.63 eV for ZZSe-NW) (Supplementary Fig. 19). The etching rate is presented using a polar plot (Fig. 4e), confirming that the three equivalent ZZSe-NW edges have lower etching rate than ZZMo-NW edges. Using the etching rate from the polar plot and an initial parallelogram pore, the simulated shape evolution follows a truncated parallelogram, a truncated triangle, and finally a triangle due to the much faster disappearance of the two ZZMo-NW edges (Fig. 4f). These findings match experimentally observed shape evolution at every stage (Fig. 4a–d).

In summary, in situ STEM imaging, first principles theory, and AIMD simulations were used to understand the evolution of edge structure around pores in $Mo_{1-x}W_xSe_2$ monolayers as a function of temperature and Mo chemical potential. The results indicate the potential of atomistic imaging of 2D materials and in situ STEM as platforms to understand and direct synthesis by control over the thermal and chemical environments to measure growth kinetics, understand growth mechanisms, and develop predictive models for both suspended and on-substrate 2D materials. These results indicate the importance of adsorbates on altering synthesis pathways, and provide opportunities for further explorations of tailoring chemical reactions through intentional introduction of reactants with the development of novel TEM-based platforms. The ability to synthesize pores with metastable NW edges having predictable atomic structures opens the door to a wide range of novel 1D-2D nano hetero-structures (e.g., semiconducting monolayers terminated or surrounded by metallic NWs) with electrical and magnetic properties as revealed by DFT, which could potentially act as functional building blocks for next-generation nano-devices.

## Methods

**Materials growth**. The $Mo_{1-x}W_xSe_2$ monolayers were synthesized with a CVD method in a tube furnace equipped with a 2" quartz tube. In a typical run, the growth substrates—e.g., Si with 250 nm $SiO_2$ ($SiO_2$/Si)—were cleaned by acetone and isopropanol (IPA) and placed face-down above an alumina crucible containing ~0.2 g of $MoO_3$ powder (for the growth of $Mo_{1-x}W_xSe_2$, a mixture of $MoO_3$ and $WO_3$ powder was used), which was then inserted into the center of the quartz tube. Another crucible containing ~1.2 g Se powder was located at the upstream side of the tube. After evacuating the tube to ~5 × 10⁻³ Torr, 40 sccm (standard cubic centimeter per minute) argon and 6 sccm hydrogen gas were introduced into the tube, and the reaction was conducted at 780 °C (with a ramp rate of 30 °C/min) for 5 min at a pressure of 20 Torr. At 780 °C, the temperature at the location of Se powder was ~290 °C. After growth, the furnace was allowed to cool naturally to room temperature.

**In situ scanning transmission electron microscopy**. A commercially available Protochips Fusion in situ heating platform from Protochips Inc. (Morrisville, NC) was used for the in situ heating experiments within an aberration-corrected Nion UltraSTEM100, operating at 100kV with a spatial resolution of ~1Å. The UltraSTEM is equipped with a probe aberration corrector and was used to acquire HAADF-STEM images using a convergence angle of 31 mrad and a collection angle between 86 and 200 mrad. The microfabricated e-chip heating platforms contain a thin film SiC heating element to resistively heat specimens. Each e-chip has a 3 × 3 array of SiN membrane windows. On each window, focused ion beam milling was used to create holes with roughly 0.4 μm diameter through which materials transformation can be observed without overlapping with SiN. The monolayer samples were prepared using a wet transfer process. Poly(methyl methacrylate) (PMMA) was first spun onto the $SiO_2$/Si substrate with monolayer crystals at 3500 r.p.m. for 60 s. The PMMA-coated substrate was then floated on 1 M KOH solution, which etched the silica epi-layer, leaving the PMMA film with the monolayer crystals floating on the solution surface. The film was then rinsed several times on a glass with deionized water to remove residual KOH. The washed film was transferred onto the Protochips heating chip in the following way: The chip was put into a funnel filled with deionized water followed by transfer of the floating PMMA/$MoSe_2$ film from a beaker with water to the funnel using a glass slide. Then the water was slowly drained from the funnel while guiding the floating film toward the chip at the bottom of the funnel where transfer finally occurred. Such a process ensures placement of monolayers sitting on the window portion of the microchips for observation. The PMMA was then removed with acetone and the samples were then soaked in methanol for 12 h to achieve a clean monolayer surface (some regions with relative high amount of polymer residues also exist as shown in the main text). STEM images were simulated using the code developed by Kirkland[39].

**Density functional theory calculations**. DFT calculations were performed with the Vienna Ab initio Simulation Package (VASP)[36]. The exchange-correlation functional was described by the Perdew-Burke-Ernzerhof version of generalized gradient approximation[40] and the core region by the projector augmented wave method[41]. Spin-polarized calculations are performed. The Brillouin zone was sampled using a 1 × 19 × 1, 1 × 11 × 1, 1 × 9 × 1, and 1 × 7 × 1 Monkhorst-Pack k-point meshes for the 1 × 1, 1 × 2, 1 × 3, and 1 × 4 supercells, respectively, to relax the structures. Finer k-point meshes were used for LDOS calculations and Gaussian smearing with a sigma value of 0.01 eV was utilized. The calculated lattice constant of 3.32 Å for the $MoSe_2$ monolayer is used to set up the supercells. Vacuum spaces in x and z directions are larger than 10 Å. The plane-wave cutoff is set to 300 eV, and a conjugate gradient method is applied to relax the geometry until interatomic forces are less than 0.01 eV/Å. No substrate is involved in the calculations, as the observed monolayer regions in the in situ STEM heating experiments were suspended over holes. The density of k-point sampling and value of cutoff energy are tested to be sufficient.

**Ab initio molecular dynamics simulation**. The AIMD simulations were also performed using VASP. Considering the time-consuming nature of the AIMD, the plane-wave cutoff is set to be 184 eV and the Brillouin zone is sampled at the Γ point only. All AIMD trajectories were run with the canonical ensemble (NVT) using a Nosé thermostat[42] at a temperature from 1500 to 1000 K. The time step of the AIMD simulation is 1 fs. The total time scales of the trajectories are more than 100 picoseconds (ps). To simulate the effect of e-beam irradiation, Se atoms near the edge were artificially removed at the beginning of AIMD simulation and the leftover Mo and Se atoms at the edge were allowed to gradually aggregate into clusters, chains, and eventually to form a stable nanowire. The other side of the structure is fixed.

**Data availability**. The data that support the findings of this study are available from the corresponding authors upon request.

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

## Acknowledgements

In situ aberration corrected STEM experiments were conducted at Oak Ridge National Laboratory's Center for Nanophase Materials Sciences (CNMS), a U.S. Department of Energy Office of Science User Facility. (X.S. and R.R.U.) Synthesis science sponsored by the Materials Science and Engineering Division, Office of Basic Energy Sciences, U.S. Department of Energy. (X.L., C.M.R., D.G., K.X.) This research used resources of the National Energy Research Scientific Computing Center, a DOE Office of Science User Facility supported by the Office of Science of the U.S. Department of Energy under Contract No. DE-AC02-05CH11231. W.Z., J.D., and F.D. acknowledge the support from the Institute for Basic Science (IBS-R019-D1) of Korea.

## Author contributions

X.S., X.L., F.D., K.X., R.R.U. conceived the idea. X.S. and R.R.U. designed and performed the in situ STEM experiments, and analyzed the data. X.L., C.M.R., D.B.G., and K.X. synthesized the $MoWSe_2$ samples. Z.W., J.D., and F.D. performed DFT and MD simulations. X.S., X.L., and W.Z. drafted the manuscript. All authors contributed to data analysis, result interpretation and writing the manuscript.

## Additional information

**Competing interests:** The authors declare no competing interests.

