## [Peer Review File · Nature Communications]

Reviewers' Comments:

Reviewer #1:

Remarks to the Author:

The authors present an in situ study of the edge evolution under controlled conditions of temperature and chemical potential in 2D transition metal dichalcogenides. Using real-time STEM imaging, different kind of edges containing Se termination or nanowires along the edge were observed. The authors performed molecular dynamics simulations to determine the edge evolution pathways and conditions for structural stability, and obtained the same structures as observed by STEM. They also used DFT calculations to gain insight on the electronic structure of the different edges. Although in reference 26, similar kind of edges were observed, in that work the focus was more on the free standing nanowires. In the present work these edges are studied in great details. This is a beautiful study that demonstrate the power of combining in situ structural characterization techniques with ab initio methods in order to understand the dynamics of formation of low-dimensional systems. I do appreciate the importance of understanding the formation and evolution of different edges configurations in 2D materials as well as their electronic structures. My only concern is that this study is on suspended films. I wonder if the presence of a substrate will affect the energetics of the entire process, since (to the best of my knowledge) those kind of nanowire-edge have not been observed in substrate-supported 2D films. If that is the case, the potential impact of this study on device applications, as claimed by the authors, could be somehow limited. Other than that, this is an original and excellent piece of work with high scientific standards that deserve publication in a highly reputed journal, I do recommend its publication in its present form.

Reviewer #2:

Remarks to the Author:

The authors used high resolution, in situ TEM to probe the edge reconstructions in MoxWxSe_{1-x} type 2D systems. They further showed that the types of edge structures influenced the development of subsequent etch patterns.

The ability of the authors to capture in situ STEM images in changing chemical environment is impressive. The shape evolution of pores from parallelogram to triangle shape is convincingly presented. Overall the data is nicely presented, the paper is well written, the analysis is solid and it deserves to be published in Nature Communications.

Below are some comments for improvements if possible.

1) I am not sure why a ternary system is chosen in this study, any particular purpose, and how that will differ from a binary one in terms of the edge reconstructions, do we have a larger diversity of edge structures for a ternary system ?

2) The implication of the study – what does it mean practically, what message can we draw and how do we apply ? This is the question. If we have a collection of observations here, what is the key central message for CVD people how to grow what ?

Is there possibility to control these structures, meaning getting a high density and homogeneous termination of these? While it is nice to see the structures with clarity at the edges of pores, what can we do with these ?

3) the authors mentioned that Mo can be consumed by C to form Molybdenum carbide and that changes the chemical potential, i.e. reduces chemical potential of Mo. Is this observation a specialised case restricted to the in situ TEM environment, or can it be generalised to actual CVD

growth ?

Literatures on the growth of 1D MoSe₂ wires as well as their edge structures can be more complete, below are some past work which have analysed edge structures of 1D MoSe₂. I suggest the authors cite them to reflect the fact that increasing attention is now being paid to 1-D TMD structures, even when these occur at the discontinuity of 2D structures, eg. domain boundaries or edges. Otherwise the motivation of studying all these pore structure is not so clear.

References

1. Controlled Growth of 1D MoSe₂ Nanoribbons with Spatially Modulated Edge States
Fang Cheng, et. al. Nano Lett., 17 (2), 1116–1120 (2017)

2. Large Area Synthesis of 1D-MoSe₂ Using Molecular Beam Epitaxy
Sock Mui Poh et. al., Advanced Materials, 2017, 1605641 (2017)

3. Oscillating Edge States in One-dimensional MoS₂ Nanowires
Xu hai and Kian Ping Loh* et al, Nature Communications, 7, 12904 (2016)

Reviewer #3:

Remarks to the Author:

This report is a product of collaboration between well-equipped and skillful experimental lab, capable to directly image the edges voids in TMD Mo_{1-x}W_xSe₂ layers via atom-resolved STEM, and theorists versed in DFT tools to compute the energy of various structures (emerging under the beam). This allows one to compare the energies of some metastable atomic configurations at assumed chemical potentials of the elemental species, and explain some of the observed trends. Both experiment and theory components are well executed but reveal no surprises in scientific content, I admire the images but learn little beyond details. Pretty much paper follows the "observecompute/explain" mode. I do not see any "predictive" modeling (as abstract says) or surprising observations. In this sense the good quality report is for materials science professionals, not really of broad interest. There is also no directed engineering, no really controllable targeted change to desired property, and no property characterization of "desired functionality" as the abstract promises. The diagrams in 2e-f are meaningful but can now be routinely done, years ago for particles (not holes, which makes no difference), the authors should be aware of old papers (and many more recent in "beyond-graphene" era) where such calculations were nicely performed and similarly plotted, a few to mention

Georg Kresse (yes, the VASP one) and co-authors "Shape and edge sites modifications of MoS₂ catalytic nanoparticles induced by working conditions: A theoretical Study". J. Catal. 2002, 207, 76–87.

Nørskov -Helveg work in Phys. Rev. Lett. 2001, 87, No. 196803.

Li, et al "MoS₂ nanoribbons: High stability and unusual electronic and magnetic properties" JACS 2008, 130, 16739–16744.

Role of residual carbon qualitatively is clear, to deplete Mo makes sense, but must be too obscure to quantify it or to rely on it as control parameter for the engineering.

Overall paper has many interesting, quality images and may fit into good materials journal, perhaps Adv Mater. or 2D Mater. I cannot recommend it to NatComm.

Response to Reviewers' Comments

We would like to thank the reviewers for their positive comments and useful suggestions.

Reviewer #1 (Remarks to the Author):

The authors present an in situ study of the edge evolution under controlled conditions of temperature and chemical potential in 2D transition metal dichalcogenides. Using real-time STEM imaging, different kind of edges containing Se termination or nanowires along the edge were observed. The authors performed molecular dynamics simulations to determine the edge evolution pathways and conditions for structural stability, and obtained the same structures as observed by STEM. They also used DFT calculations to gain insight on the electronic structure of the different edges. Although in reference 26, similar kind of edges were observed, in that work the focus was more on the free-standing nanowires. In the present work, these edges are studied in great details. This is a beautiful study that demonstrate the power of combining in situ structural characterization techniques with ab initio methods in order to understand the dynamics of formation of low-dimensional systems. I do appreciate the importance of understanding the formation and evolution of different edges configurations in 2D materials as well as their electronic structures. My only concern is that this study is on suspended films. I wonder if the presence of a substrate will affect the energetics of the entire process, since (to the best of my knowledge) those kind of nanowire-edge have not been observed in substrate-supported 2D films. If that is the case, the potential impact of this study on device applications, as claimed by the authors, could be somehow limited. Other than that, this is an original and excellent piece of work with high scientific standards that deserve publication in a highly reputed journal, I do recommend its publication in its present form.

R. We thank the reviewer for the comments on our work. As for the influence of the substrate, we agree with the reviewer that the substrate may influence the energetics of NW-terminated edge formation. One could speculate that the diffusion barrier may be lowered at elevated temperatures, which may allow for similar atomic structure transformations to be attained on a substrate. However, this would need to be substantiated in future studies, where the complex

interplay between the structure and chemistry of the substrate and 2D transition metal dichalcogenides needs to be well understood and the chemical potential that mediates the formation of different types of edges needs to be controlled. Recently, by using molecular beam epitaxy with precisely controlled chemical potentials, the various edges including 1D Mo_6Te_6 NWs/2D MoTe_2 monolayer structure, similar to the observed edges in our work, have been reported on substrates (Yu, et al, Nano Lett., 2018, 18 (2), pp 675–681; Zhao, et al, Nano Lett., 2018, 18 (1), pp 482–490), which further verify the feasibility of the growth of these NW-based edges on substrates.

Reviewer #2 (Remarks to the Author):

The authors used high resolution, in situ TEM to probe the edge reconstructions in MoxWxSe_{1-x} type 2D systems. They further showed that the types of edge structures influenced the development of subsequent etch patterns.

The ability of the authors to capture in situ STEM images in changing chemical environment is impressive. The shape evolution of pores from parallelogram to triangle shape is convincingly presented Overall the data is nicely presented, the paper is well written, the analysis is solid and it deserves to be published in Nature Communications.

R. We thank the reviewer for the comments.

Below are some comments for improvements if possible.

1) I am not sure why a ternary system is chosen in this study, any particular purpose, and how that will differ from a binary one in terms of the edge reconstructions, do we have a larger diversity of edge structures for a ternary system?

R. One of the reasons why we chose a ternary system is because the bright W dopants (with low concentration) can be used as fiducial markers to better align sequentially acquired STEM images. W dopants also help to understand the diffusion of atoms during dynamic process. For example, as shown in the figure below, we can see how the three W dopants indicated by white arrows do not move during the edge formation while two Mo atoms indicated by red arrows move to form the nanowire. This is very helpful for understanding the dynamic restructuring

process at the atomic scale. W dopants are also very useful when determining the edge orientation when Mo sites and 2Se sites are difficult to distinguish due to close contrast. From our experiments, the low-concentration W dopants do not appear to influence the edge construction in the pure MoSe₂ region. However, we cannot rule out the possibility that the metal substituents provide an additional way in which the edge structure might be altered.

Figure R1. Edge evolution as revealed in Figure 3c-d in the main manuscript. The W dopants indicated by white arrows remain at their positions between the two frames, while the W dopants indicated by red arrows have moved to form the nanowire.

2) The implication of the study – what does it mean practically, what message can we draw and how do we apply? This is the question. If we have a collection of observations here, what is the key central message for CVD people how to grow what?

R. The key message of this paper is that by locally tuning the chemical environment in 2D materials, we can create different edge structures in 2D TMDs. The evolution processes are predicted by DFT simulation, which can be used as a predictive tool for the controlled synthesis of 2D TMD with engineered interfacial edge structures tailored for functional applications. During CVD, if we can precisely tune the chemical potentials of metal and chalcogen of TMDs by controlling the growth kinetics and chemical environment during growth, the theoretically-designed, specific edges can be controllably grown on substrates for various applications. For example, by switching the growth or etching process, we could change the local chemical

environment to create different edges to form different junctions 1D metal/semiconductor-2D semiconductor, even magnetic edges which are potentially useful for electronics, quantum emitters and catalysis. However, to realize this requires more detailed studies in the future, including an in depth understanding of the complex interplay between the structure and chemistry of the substrate and 2D TMDs and controlling the chemical potential that mediates the formation of different types of edges.

Is there possibility to control these structures, meaning getting a high density and homogeneous termination of these?

R. We thank the reviewer for this great comment. The key to control these structure is that how we can precisely control the chemical potentials during the growth/etching process. In our work, actually there is some difficulty to precisely control the local chemical environments in the STEM chamber to attain a homogeneous termination of specific type of edge structures. However, it is possible to get a high density and homogeneous terminations of these structure using some methods which could precisely control the chemical potentials such as molecular beam epitaxy or pulsed laser deposition. Recently, by using molecular beam epitaxy with precisely controlled chemical potentials, the various terminated edges including the 1D Mo_6Te_6 NWs/2D MoTe_2 monolayer structure, similar to the observed edges in our work, have been reported on substrates (Yu, et al, Nano Lett., 2018, 18 (2), pp 675–681; Zhao, et al, Nano Lett., 2018, 18 (1), pp 482–490.).

While it is nice to see the structures with clarity at the edges of pores, what can we do with these?

R. By using this nanoporous MoSe_2 films with different terminated edges, there may exist some novel functional applications from catalytic and magnetic properties at edge sites, as demonstrated in recent work (Zhao, et al, Nano Lett., 2018, 18 (1), pp 482–490). Also, controlling edges on nanopores of suspended 2D membranes is in itself highly important. Compared to 2D flakes on substrate, the nanopores and edges of suspended flakes have additional applications such as water desalination (Nature Communications volume 6, 2015, 8616) and single molecular detection (Chem. Soc. Rev., 2016,45, 476-493).

3) the authors mentioned that Mo can be consumed by C to form Molybdenum carbide and that changes the chemical potential, i.e. reduces chemical potential of Mo. Is this observation a specialised case restricted to the in situ TEM environment, or can it be generalised to actual CVD growth?

R. This is probably a general case as Mo and C have a strong affinity to bond together. We also observe such phenomenon when MoS₂ flake was heated to high temperature on an amorphous carbon substrate. In our *in situ* heating experiment in STEM, carbon from the polymer residues on the monolayers after transfer, serves perfectly as an agent to control the Mo chemical potential since they tend to bond together. However, in the real CVD system, there are more ways to control the chemical environment, i.e., the precursor, the gas flow, the temperature, and the chemical or physical properties of the substrate. Of course, it is feasible to intentionally introduce carbon sources, e.g., carbon-containing gases or amorphous carbons, with proper amount.

Literatures on the growth of 1D MoSe₂ wires as well as their edge structures can be more complete, below are some past work which have analysed edge structures of 1D MoSe₂. I suggest the authors cite them to reflect the fact that increasing attention is now being paid to 1-D TMD structures, even when these occur at the discontinuity of 2D structures, eg. domain boundaries or edges. Otherwise the motivation of studying all these pore structure is not so clear.

References

*1. Controlled Growth of 1D MoSe₂ Nanoribbons with Spatially Modulated Edge States
Fang Cheng, et. al. Nano Lett., 17 (2), 1116–1120 (2017)*

*2. Large Area Synthesis of 1D-MoSe₂ Using Molecular Beam Epitaxy
Sock Mui Poh et. al., Advanced Materials, 2017, 1605641 (2017)*

3. Oscillating Edge States in One-dimensional MoS₂ Nanowires

Xu hai and Kian Ping Loh et al, Nature Communications, 7, 12904 (2016)*

R. We thank the reviewer to provide relevant references and have cited them in the introduction part to motivate the study.

Reviewer #3 (Remarks to the Author):

This report is a product of collaboration between well-equipped and skillful experimental lab, capable to directly image the edges voids in TMD $Mo_{1-x}W_xSe_2$ layers via atom-resolved STEM, and theorists versed in DFT tools to compute the energy of various structures (emerging under the beam). This allows one to compare the energies of some metastable atomic configurations at assumed chemical potentials of the elemental species, and explain some of the observed trends. Both experiment and theory components are well executed but reveal no surprises in scientific content, I admire the images but learn little beyond details. Pretty much paper follows the "observecompute/explain" mode. I do not see any "predictive" modeling (as abstract says) or surprising observations. In this sense, the good quality report is for materials science professionals, not really of broad interest. There is also no directed engineering, no really controllable targeted change to desired property, and no property characterization of "desired functionality" as the abstract promises.

R. We would like to thank the reviewer for the comments on our study. We agree with the reviewer that the current manuscript is written in a way that follows the observe→computer/explain mode. A more proper statement would be along the lines that the excellent match between experiment and theory in this paper paves the way for predictive study on future 2D material systems. In the abstract, we have moved "predictive" that originally described our work to after "provide a pathway" to describe future works.

The diagrams in 2e-f are meaningful but can now be routinely done, years ago for particles (not holes, which makes no difference), the authors should be aware of old papers (and many more recent in "beyond-graphene" era) where such calculations were nicely performed and similarly plotted, a few to mention

Georg Kresse (yes, the VASP one) and co-authors "Shape and edge sites modifications of MoS2 catalytic nanoparticles induced by working conditions: A theoretical Study". J. Catal. 2002, 207, 76–87.

Nørskov -Helveg work in Phys. Rev. Lett. 2001, 87, No. 196803.

Li, et al "MoS2 nanoribbons: High stability and unusual electronic and magnetic properties" JACS 2008, 130, 16739–16744.

R. We would like to thank the reviewer for bringing up these relevant references on density functional theory (DFT) simulation of edge structures in 2D materials. We have cited these references in the revised version. However, existence of such DFT tools should not undermine the novelty of our paper. The DFT simulation was utilized here to understand the *in situ* experiment with details about edge structures that were never reported before: real time atomic resolution observation of edge structural evolution depending on the chemical environment, and the first observation of NW-terminated edges that form a 1D-2D heterostructures with potentially unique properties.

Role of residual carbon qualitatively is clear, to deplete Mo makes sense, but must be too obscure to quantify it or to rely on it as control parameter for the engineering.

Overall paper has many interesting, quality images and may fit into good materials journal, perhaps Adv Mater. or 2D Mater. I cannot recommend it to NatComm.

R. We agree with the reviewer that it might be difficult to quantify or control the residual carbon for edge engineering as the carbon residue is randomly distributed on the MoWSe₂ flakes. However, not limited to carbon residue, some other adsorbents and gaseous chemicals may also be used to control the Mo chemical potential, which is the key parameter here that controls the edge structure. The importance of our paper is not to demonstrate how precisely engineer the edge structure using residual carbon, but to demonstrate that these edges can be structurally transformed to theoretically predicted metastable atomic configurations by thermal and chemical driving forces. Therefore, this work provides a pathway for the predictive and controlled synthesis of edge configurations in 2D materials to achieve desired functionality.